# Long-Term Outcomes of Service Women Injured on Combat Deployment

**DOI:** 10.3390/ijerph18010039

**Published:** 2020-12-23

**Authors:** Jessica R. Watrous, Cameron T. McCabe, Amber L. Dougherty, Abigail M. Yablonsky, Gretchen Jones, Judith Harbertson, Michael R. Galarneau

**Affiliations:** 1Naval Health Research Center, San Diego, CA 92152, USA; cameron.t.mccabe.ctr@mail.mil (C.T.M.); amber.l.dougherty.ctr@mail.mil (A.L.D.); abigail.m.yablonsky.mil@mail.mil (A.M.Y.); gretchen.m.jones6.ctr@mail.mil (G.J.); judith.harbertson.ctr@mail.mil (J.H.); michael.r.galarneau.civ@mail.mil (M.R.G.); 2Leidos, Inc., San Diego, CA 92152, USA; 3Naval Medical Center San Diego, San Diego, CA 92134, USA; 4Axiom Corporation, San Diego, CA 92152, USA

**Keywords:** women’s health, military, veterans, mental health, patient-reported outcomes, health behaviors

## Abstract

Sex- and gender-based health disparities are well established and may be of particular concern for service women. Given that injured service members are at high risk of adverse mental and behavioral health outcomes, it is important to address any such disparities in this group, especially in regard to patient-reported outcomes, as much of the existing research has focused on objective medical records. The current study addressed physical and mental health-related quality of life, mental health symptoms, and health behaviors (i.e., alcohol use, sleep, and physical activity) among a sample of service women injured on deployment. Results indicate that about half of injured service women screened positive for a mental health condition, and also evidenced risky health behaviors including problematic drinking, poor sleep, and physical inactivity. Many of the mental and behavioral health variables demonstrated statistically significant associations with each other, supporting the relationships between psychological health and behaviors. Results provide additional evidence for the importance of access to integrated and effective mental healthcare treatment for injured service women and the need for screening in healthcare settings that address the multiple factors (e.g., mental health symptoms, alcohol use, poor sleep) that may lead to poor outcomes.

## 1. Introduction

Sex- and gender-based disparities in knowledge, access to care, and health outcomes are well-documented, and extant research suggests that, although biologically based disparities exist (e.g., cervical cancer rates), many are related to nonbiological (e.g., social) determinants [1]. Despite direct efforts to address these inequalities through research (e.g., National Institutes of Health’s required inclusion of women) [2] and clinical initiatives (e.g., raising awareness about cardiovascular disease among women [3]), disparities persist between men and women in research, patient safety, person-centered care, effective treatment, and healthy living [4].

Knowledge and care accessibility disparities are particularly relevant to military women given that clear deficits in women’s health research exist [5,6,7] and the military heath system was established prior to significant increases in the number of women serving in the military. As part of their scoping review on the health of military women, Yablonsky and colleagues noted that, since the Selective Service Act expired in 1973, the percentage of women serving in the military has increased over time [8]; today, women represent 16.5% of the total military force [9]. The researchers identified multiple knowledge gaps in the current body of health research among service women in the following areas: psychological health, readiness, acute care and preventive medicine, deployment health, social relationships, chronic illness, and obstetric–gynecologic health [6,8]. Additionally, a recent report by the Defense Health Board on Active Duty Women’s Health Care Services underscored the importance of women to the success of the military and noted that despite this value, military women continue to experience well-documented health disparities for which previous recommendations have not lead to long-term improvements [10].

Even before service women were allowed to serve in direct ground combat positions in 2015, they were at risk of injury in the Iraq and Afghanistan conflicts given the asymmetric (i.e., indirect) nature of the warfare [11,12]. The risk of injury is particularly important to consider in the healthcare of military women as injured service members face a heightened possibility of developing physical and mental health problems in their lifetime, with negative impacts on their quality of life [13,14,15]. Extant research indicates that those injured in war often experience mental and behavioral health sequelae following their injury, particularly posttraumatic stress disorder (PTSD) and depression [13,15], and are likely to engage in risky health behaviors such as hazardous alcohol use and demonstrate poor engagement in health promoting behaviors such restful sleep and physical activity [16].

To date, more than 1000 service women have been wounded in action in the Iraq and Afghanistan conflicts, as documented by the Defense Casualty Analysis System, a Department of Defense database that aggregates data to provide counts of U.S. warfighters injured or killed in U.S. military operations [17]. Research on this group of service women is sparse and, consistent with the military injury literature overall, primarily focuses on objective medical outcomes [12,18]. Much less is known about patient-reported outcomes of military women following physical injury, including mental health symptom severity, health behaviors, and physical and mental health-related quality of life (HRQOL). Thus, research describing these patient-reported outcomes for service women injured on deployment is a necessary first step in identifying the potential adverse outcomes these women may be experiencing, lines of future research, and areas of clinical care improvements. Understanding functioning and wellness in multiple domains is particularly important for injured service members given that multimorbidity is associated with worsened quality of life [19] and likely contributes to the over 700 billion U.S. dollars estimated to be spent on healthcare for this generation’s warfighters [20].

The current study addresses knowledge gaps related to patient-reported outcomes for military women injured on deployment who are participating in the Wounded Warrior Recovery Project (WWRP). Specifically, the current study addresses (1) demographic, service, and injury-related characteristics; (2) mental and behavioral health patient-reported outcomes; and (3) associations between these variables of interest.

## 2. Materials and Methods

### 2.1. Participants

Participants in the WWRP, a long-term surveillance project of patient-reported outcomes of service members from all military branches who were injured during combat deployment, were included in the current study. Participants were included if they were service women injured between March 2002 and August 2014 who completed mental health (i.e., PTSD and depression) and health behavior assessments between September 2018 and April 2020. About 5% of the overall WWRP sample (*n* = 6296 as of December 2020) are women. The final study sample consisted of 230 service women.

### 2.2. Procedure

All study procedures were approved by the Naval Health Research Center (NHRC) Institutional Review Board. Previous publications have detailed the WWRP method [21]. Briefly, participants are identified via the Expeditionary Medical Encounters Database (EMED), an NHRC-maintained dataset of deployment-related clinical encounters [22]. Individuals are eligible to participate in WWRP if they received care for an injury during an overseas contingency operation (e.g., Operation Iraqi Freedom, Operation Enduring Freedom) and are not currently deployed at the time of a given WWRP recruitment effort. Recruitment is conducted on a rolling basis and is ongoing. Individuals are contacted via postal mail and email and offered the opportunity to enroll in WWRP via the online survey portal. Participants complete study informed consent materials prior to completing a baseline assessment. Participants are then asked to complete an assessment every 6 months for 15 years and are compensated with a $20 (USD) Amazon gift card for each assessment iteration they complete. The WWRP team utilizes multiple methods to remind participants when follow-up assessments are due, including text messages and phone calls. As of August 2018, participants are asked to complete assessments of health behaviors (e.g., smoking, alcohol use, physical activity) at each follow-up assessment. The current study focuses on the first health behavior assessments that participants completed after these were launched, as well as mental health and HRQOL measures that were completed during the same assessment timepoint.

### 2.3. Measures

#### 2.3.1. Demographic, Service, and Injury-Related Characteristics

Demographic and service-related variables at the time of injury were extracted from the Defense Manpower Data Center. Demographic variables included sex, age, and race/ethnicity. Service-related variables included service branch (Air Force, Army, Marine Corps, or Navy) and rank (enlisted or officer/warrant officer). EMED data provided comprehensive injury information including Injury Severity Score (ISS), injury mechanism, and number of years since injury. ISS, which is derived from the Abbreviated Injury Scale [23], indicates overall trauma severity with a focus on mortality risk [24], and is useful for quantifying the impact of multiple injuries. The AIS classifies the severity of each injury by 9 body regions on a scale from 1 (minor) to 6 (currently untreatable). The ISS is derived from the sum of squares of the highest AIS code in each of the 3 most severely injured body regions and ranges from 0 (no injury) to 75 (fatal injury). Injury mechanism was divided into 3 primary categories: blast, gunshot, or other/unknown.

#### 2.3.2. Health-Related Quality of Life (HRQOL)

A modified 36-item Short Form Survey (SF-36) was used to assess HRQOL among participants [25]. The SF-36 is intended to address key health concepts including functioning, pain, limitations due to physical and mental health problems, overall health rating, and change in health status [26]. The SF-36 has been shown to have sound psychometric properties, including convergent and discriminant validity [27], and has been commonly used to assess quality of life in military research [28,29]. Physical (PCS) and mental health component scores (MCS) were calculated on the basis of normative scoring procedures from Ware et al. [30], with scores ranging from 0–100, and with higher scores indicative of higher HRQOL.

#### 2.3.3. PTSD Symptoms

The 20-item PTSD Checklist for the DSM-5 (Diagnostic and Statistical Manual of Mental Disorders, 5th Edition) (PCL-5) was used to assess PTSD symptoms in the past month [31]. Participants rated, on a scale of 0 (*not at all*) to 4 (*extremely*), how bothered they were by a given symptom. Ratings were summed, with higher total scores demonstrating worse symptom severity (⍺ = 0.97), and scores greater than 33 indicated a participant screened positive for PTSD [32]. The PCL-5 has strong psychometric properties with regard to internal consistency, test–retest reliability, convergent and discriminant validity, and diagnostic utility within military personnel [33,34].

#### 2.3.4. Depression Symptoms

The 8-item Patient Health Questionnaire (PHQ-8) was used to assess depressive symptoms in the past 2 weeks. Participants reported how frequently they were bothered by symptoms using a scale of 0 (*not at all*) to 3 (*nearly every day*). Ratings were summed, with greater scores representing greater severity of depression symptoms (⍺ = 0.91), and with scores 10 or greater indicating a participant screened positive for a major depressive episode [35]. The PHQ-8 is well validated to detect and monitor depression [36].

#### 2.3.5. Alcohol Use and Problems

Alcohol use in the past 30 days was assessed with items from the National Institutes of Alcohol Abuse and Alcoholism (NIAAA) [37]. Participants were presented with the NIAAA’s graphic that defines standard drinks (i.e., 12 oz. of beer, 8–9 oz. of malt liquor, 5 oz. of wine, 1.5 oz. of liquor; NIAAA, 2019) and asked to report (1) on how many days they consumed alcohol in the past month (frequency); (2) how many alcoholic drinks they consumed on a typical drinking day (quantity); and (3) whether they engaged in heavy episodic drinking (HED), defined as consuming 4 or more drinks within a 2-h period for women.

The 10-item Alcohol Use Disorders Identification Test (AUDIT) was used to assess alcohol-related problems [38]. Participants reported how frequently they experienced problems consistent with hazardous drinking in the past year. Scores of 8 or greater reflected hazardous alcohol use (⍺ = 0.86). The AUDIT has strong reliability and validity in military and civilian samples [39,40].

#### 2.3.6. Sleep

The 7-item Pittsburgh Sleep Quality Index (PSQI-7) was used to calculate three sleep-related outcomes [41]. The PSQI is a frequently used self-reported sleep measure with strong reliability and validity among civilian and military populations [42,43,44]. Sleep quality (*How would you rate your sleep quality overall?)* was measured using a scale from 1 (*very bad*) to 4 (*very good*). Typical sleep duration over the past 30 days was assessed using a single item (*How many hours do you think you actually slept each day?)*. Insomnia symptoms were calculated as the average of 2 items (*How often could you not get to sleep within 30 min?* and *How often did you wake up in the middle of the night or early morning*)? Participants responded using a scale from 1 (*never*) to 4 (*3+ times per week*).

#### 2.3.7. Physical Activity

Level of physical activity was assessed using the Rapid Assessment of Physical Activity (RAPA) [45]. The RAPA has demonstrated strong criterion validity in assessing physical activity among older adults [45], and has been utilized among other populations, such as military veterans with and without a history of traumatic brain injury [46]. Participants respond (yes or no) to seven items measuring level and frequency of aerobic activities and were shown a visual prompt describing types of light (e.g., leisurely walking), moderate (e.g., fast walking), and vigorous activities (e.g., running). Participants were considered active if they participated in at least 30 min of moderate exercise on 5 or more days a week, or 20 min of vigorous exercise 3 or more days a week. The RAPA also includes 2 single items assessing whether the individual engages in strength and flexibility activities at least once per week (yes/no).

#### 2.3.8. Statistical Analysis

Analyses were conducted using IBM SPSS Statistics v25 (IBM, Armonk, NY, USA) [47]. Descriptive statistics in the form of percentages and measures of central tendency are displayed in Table 1 and Table 2. Lastly, bivariate correlations were calculated and are presented in Table 3. To account for the number of correlations tested and control for type I error, we applied the Benjamini–Hochberg procedure using a conservative false discovery rate of 0.10 [48] (Benjamini and Hochberg, 1995). Pearson correlations are presented below the diagonal, and Spearman’s rho is presented above the diagonal.

## 3. Results

### 3.1. Demographic, Service, and Injury-Related Characteristics

Table 1 presents the descriptive data on the demographic, service, and injury-related variables for participants. On average, participants were approximately 38 years old when they completed the survey. Most participants were non-Hispanic, White, and unmarried. Most served in the Army or Marine Corps when injured, were enlisted, and almost half were active duty at the time of the assessment. Participants were injured about 10 years before completing the assessments, primarily injured by blasts, with an average ISS of about 3, indicating overall injury was mild to moderately in severity.

### 3.2. Mental and Behavioral Health

Table 2 presents descriptive statistics for HRQOL (MCS and PCS) metrics and all mental and behavioral health variables included. Overall, 49% of the sample screened positive for any mental health problem, 45% screened positive for depression, 40% screened positive for PTSD, and a majority (65%) screened positive for both PTSD and depression. Of those who provided health behavior data, 72% drank alcohol on 1 or more days in the past month. Subsequent descriptive analyses excluded participants who did not report drinking alcohol during the past 30 days. Past month drinkers drank about 2.4 (SD = 1.8) standard drinks on drinking days, 29% reported HED, and 19% met criteria for hazardous alcohol use. In regard to sleep, a majority of the sample (81%) reported getting less than the minimum recommendation of 7–9 h of sleep per night, on average. Most participants (65%) rated their sleep quality as bad, and 58% reported insomnia symptoms three or more times per week. For physical activity, about 54% of participants were under-active on the basis of responses to aerobic items on the RAPA. Forty six percent and 47% of participants engaged in strength-building or activities to increase flexibility, respectively.

Table 3 presents bivariate correlations among study variables. All correlations remained statistically significant after applying the Benjamini–Hochberg procedure. Physical and mental HRQOL, as measured by PCS and MCS scores, were significantly and negatively associated with major depression symptoms, PTSD symptoms, and insomnia symptoms, and positively related to sleep quality, and aerobic and strength activity. In addition, MCS scores were negatively associated with hazardous alcohol use. Depression and PTSD symptoms were highly correlated (*r* = 0.82, *p* < 0.001). Both depression and PTSD symptoms were significantly and negatively associated with sleep quality, recommended sleep, and aerobic and strength activity, and were positively related to insomnia symptoms. In addition, PTSD symptoms were positively associated with HED and hazardous alcohol use. 

Frequency of past month alcohol use was positively associated with other alcohol-related factors (i.e., average drinks per drinking day, HED, and hazardous alcohol use). Similarly, average drinks per drinking day was positively associated with other alcohol-related factors (i.e., HED and hazardous alcohol use) and average sleep duration. Lastly, HED was significantly and positively associated with hazardous alcohol use.

Sleep factors were interrelated. In addition to the correlations reported previously, self-reported sleep quality was significantly and positively associated with recommended sleep, and negatively associated with insomnia symptoms. Insomnia symptoms and average sleep duration were also negatively associated with recommended sleep. Lastly, sleep quality was positively associated with aerobic activity.

Physical activity factors (aerobic, strength, and flexibility) were positively correlated.

## 4. Discussion

Historically, sex- and gender-based health disparities, including knowledge gaps, have negatively impacted the health and well-being of women [1,4]. Addressing knowledge gaps is critical to identifying potential health access and outcomes disparities given that women’s representation in the military is increasing rapidly [8] and that decades of research and recommendations have not led to long-lasting improvements in active-duty women’s health [10]. Previous research on deployment-related injuries has typically included majority male samples since most injured personnel from the current conflicts have predominantly been male service members [13,14,15]. Nevertheless, service women have been and continue to be injured in the ongoing overseas contingency operations [17]. In order to address current and future health disparities, it is important to examine injury-related sequelae, particularly concurrent mental and behavioral health factors, among service women. The current study extends previous injury research, which has typically focused on objective medical outcomes, by focusing on patient-reported outcomes for service women injured during combat deployments. The findings discussed here have important implications to operational readiness, given that almost half (45%) of our sample was active duty at the time of the study and many of the outcomes they experienced could impact their performance. Additionally, these findings extend to women who may have been medically retired due to deployment-related injury or its sequelae, as well as women veterans who have separated from the military and are receiving care via the Veterans Affairs (VA) or civilian providers.

Although the goal of the current study was not to directly compare women and men who were injured on deployment, findings from the current study indicate similarities between these groups. Regarding demographic, military, and injury characteristics, the injured service women included in the current study were similar to the larger WWRP sample, which is predominantly men, and is representative of the overall population of service members injured on deployment [15]. Congruent with previous WWRP research that included different assessment timepoints and samples of interest [16,49], rates of current PTSD positive screens were similar (40% in the current study compared to 38–43% in previous studies), as were depression-positive screens (45% in the current study compared to 43–46% in previous studies). These rates are significantly higher than rates often reported for military personnel and veterans in general, which suggests that rates of positive PTSD screens range from 13 to 16% [50] and positive depression screens range from 3 to 28% [51]. Given that the assessment measures and time periods included in the referenced studies differed from ours, comparisons should be interpreted cautiously. However, it may be that the injured service women included in our study may be more likely to be currently experiencing PTSD and depression than their non-injured military and veteran counterparts. This is particularly concerning given that extant literature has found that veteran women are at increased risk of these mental health concerns when compared with civilian women [52].

Regarding the other health behaviors examined, the rates of risky health behaviors are similar to those reported in studies of predominately male injured service members [16]. A majority (almost 75%) of the sample were past-month drinkers, and of those drinkers, about 30% engaged in heavy episodic drinking in the past month. Additionally, 20% screened positive for symptoms that indicate hazardous alcohol use, which has been correlated in prior studies with substantial productivity loss and other serious consequences due to alcohol use among service members [53]. In comparison with other military and veteran research, quantity and frequency of drinking in this sample was higher relative to active-duty service women [54]. HED appears to be of particular concern among this sample of injured service women. In the Health-Related Behaviors Survey [55], which included a younger population, 23% of active-duty service women reported HED, which is lower than the proportion reported in the current study (29%). Given that alcohol use, particularly HED, typically decreases with age after peaking in young adulthood [56], the proportion of service women in this sample engaging in HED is higher than we would expect to see in this age group given these earlier findings. Furthermore, the prevalence of HED among the overall sample included in the Health-Related Behaviors Survey was equivalent to the current study sample. This is noteworthy since the sample was predominantly men (84%) and younger (70% were 34 years or younger) than the current sample and would be expected to be more likely to engage in HED given those demographics. Furthermore, the fact that injured service men and women report similar drinking patterns is significant because women are at higher risk of experiencing problems related to drinking more acutely and at lower quantities of alcohol consumption [57], and chronic use of alcohol may elevate women’s risk for heart and liver disease, as well as certain forms of cancer [58].

Consistent with extensive research in civilian populations [59,60], sleep was a significant problem for most injured service women; a majority of our sample reported getting fewer hours of sleep than the recommended number, and over half described their sleep as “bad” and endorsed insomnia symptoms. The cause of sleep difficulties (e.g., shift work, symptom of mental health problem, infant at home) among these service women is unknown. Nevertheless, this finding is particularly concerning given that extant literature suggests poor sleep increases the risk of a host of physical and mental health problems in the general population [61,62,63], as well as in military members [64]. These sleep issues may contribute to, or compound, the risk these service women face as a function of their injury.

Consistent with previous research, there were several significant correlations between study variables as presented in Table 3. First, both physical and mental HRQOL were negatively associated with mental health symptoms and poor sleep, highlighting that these types of issues have physiological and psychological impacts. Better sleep and physical activity were both associated with better physical and mental HRQOL. Alcohol use risk factors, such as hazardous alcohol use, appeared to demonstrate clearer relationships with mental health variables, including mental HRQOL and PTSD. One potential consideration is that the negative physical impacts of hazardous drinking (e.g., poor liver functioning) may take longer to develop; thus, the immediate negative impacts of alcohol may be psychological, whereas physical functioning impairments may be more temporally distal. Finally, consistent with previous literature [64], poor sleep was associated in the expected directions with mental health symptoms.

Our findings may have clinical implications for healthcare providers. Current mental healthcare access and practices may not be sufficiently addressing many of the problems that service members and veterans, including many of the service women in our sample, are experiencing, as indicated by low treatment utilization [65]; rising suicide rates among active-duty personnel [66]; and increased suicide rates among veterans, particularly veteran women, compared with civilians [67,68]. Among our sample, 10 years following their injury, many women were experiencing mental health symptoms. Although the current study does not allow for causal inference regarding whether the deployment injury was the cause of this mental health risk, injured service women appear to be experiencing these problems at higher rates than non-injured military personnel [50,51] and civilian women [52], and at similar rates as injured service men [15,16,49]. These findings support the importance of appropriate screening (i.e., for injury, risky health behaviors, mental health symptoms, sleep patterns) to identify those women who need further evaluation aimed at identifying those that may benefit from treatment. In light of the recommendations of the Defense Health Board [10], healthcare policymakers should consider “gender-sensitive customization” practices and the utilization of technology to improve current practices. For example, providers in clinics where injured service women are likely to be seen (e.g., physical therapy, obstetrics–gynecology) could ensure that women are screened for mental and behavioral health risk factors. Relying on mHealth and telehealth options for follow-up and treatment (e.g., via “warm hand off”) of these screenings could allow for better access to integrated, holistic care, which may be beneficial given the associations between behavioral and mental health outcomes found in our study. Although the current study cannot establish the temporal nature of these relationships, research shows bidirectional links between some of the mental and behavioral health variables included in the current study such as sleep and emotions/depressive symptoms [69,70]. Thus, it is possible that some injured service women are experiencing a cycle whereby extant mental health symptoms, poor sleep, problematic alcohol use, and physical inactivity are worsened due to the concurrent issues *and* reciprocally worsen the other issues as well [71,72].

The current study did include some limitations. First, as mentioned, data were cross-sectional, and thus temporal relationships cannot be clearly defined. Second, we did not assess other traumatic events between the time of injury and assessment, with this potentially being particularly relevant for military women who are at risk for sexual and physical violence [8], which could contribute to mental health symptoms and risky health behaviors. Third, we did not conduct diagnostic interviews and can only speak to positive or negative screening status for psychological problems, which may differ from actual diagnoses. Finally, the sample was relatively small compared with similar studies among injured service men. Despite these limitations, the study has several strengths, including that, to our knowledge, it is the first examination of patient-reported outcomes focused entirely on injured service women. Our inclusion of commonly reported mental *and* behavioral health problems also contributes to an integrated view of these types of issues.

Researchers should address the limitations of the current study by conducting future studies that focus on the longitudinal relationships between the variables of interest and incorporating assessments of other events that may impact long-term mental and behavioral health. Analyses focused on identifying potential disparities in outcomes for injured service members are also a crucial next step, and these analyses should consider whether potential differences are related to biological sex, gender, or both. These analyses should also account for potential risk factors that women may be at increased risk of experiencing (e.g., sexual and physical violence). In addition to assessing patient-reported outcomes, it would be beneficial to overlay this information with medical record data to ascertain if, and to what degree, women are seeking treatment for mental and behavioral problems. Finally, future research should focus on the order in which mental or behavioral health interventions should be administered to optimize outcomes among injured service women.

## 5. Conclusions

In summary, knowledge gaps regarding the patient-reported mental and behavioral health outcomes of injured service women exist. The current study aimed to address these gaps by examining patient-reported physical and mental health-related quality of life, mental health symptoms (i.e., PTSD and depression), and health behaviors (i.e., alcohol use, sleep, and physical activity) among a group of service women injured on deployment. The sample evidenced high rates of mental health problems and risky health behaviors that may adversely impact their long-term outcomes. Finally, consistent with extant literature, there were significant associations between many of the mental and behavioral health variables suggesting complex and potentially compounding relationships between these issues. Clinical care providers for service women should implement appropriate screening for these factors and provide targeted referrals for effective, integrated mental healthcare.

## Figures and Tables

**Table 1 ijerph-18-00039-t001:** Demographic, service, and injury-specific characteristics of injured service women (*N* = 230).

Characteristic	*n* ^a^	M (*SD*) or %
**Age**	230	38.19 (*7.90*)
**Marital status**		
Separated, divorced, or widowed	23	10.00
Married	92	40.00
Unmarried	115	50.00
**Education**		
High school or equivalent	136	59.65
Some college	16	7.02
Bachelor’s degree or higher	46	20.17
Other	30	13.16
**Service branch**		
Air Force	16	6.96
Army	171	74.35
Marine Corps	31	13.48
Navy	12	5.21
**Rank**		
Enlisted	187	81.30
Officer	43	18.70
**Military status**		
Active duty	98	44.55
Separated	122	55.45
**Injury mechanism**		
Blast	160	70.18
Gunshot wound	11	4.82
Other	57	25.00
**Injury Severity Score**	228	3.17 (*3.68*)

^a^ Sample sizes may differ due to sample restrictions and missing data on study outcomes. Italic is per APA formatting.

**Table 2 ijerph-18-00039-t002:** Mental and behavioral health descriptive data of injured service women (*N* = 230).

Measure	*n* ^a^	M (*SD*) or %
**Health-related quality of life (HRQOL)**		
Physical component score (PCS)	230	44.18 (*11.14*)
Mental component score (MCS)	230	34.95 (*13.38*)
**Mental health**		
Depression symptom severity	230	9.97 (*6.30*)
Depression positive screen	230	45.22
PTSD symptom severity	230	29.61 (*20.36*)
PTSD positive screen	230	40.00
**Health behaviors**		
Typical number of drinking days	204	5.35 (*7.52*)
Average drinks per drinking day (past month) ^b,c^	147	2.44 (*1.76*)
Heavy episodic drinking (HED) ^b^	147	28.57
Hazardous alcohol use ^b^	157	19.11
Sleep quality (good)	222	35.14
Sleep duration	167	7.65 (*2.15*)
Insomnia symptoms	222	3.29 (*0.77*)
Getting recommended sleep (7–9 h)	209	18.66
Meeting physical activity recommendations		46.08
Aerobic	217	45.46
Strength	220	46.82
Flexibility	220	44.18 (*11.14*)

^a^ Sample sizes may differ due to sample restrictions and missing data on study outcomes. ^b^ Items restricted to past-month drinkers. ^c^ Two past-month drinkers reported consuming 0 drinks on their typical drinking day. Scores were recoded as 0.5, indicating a nonzero average level of consumption. Italic is per APA formatting.

**Table 3 ijerph-18-00039-t003:** Bivariate correlations between quality of life, mental health, and behavioral health variables.

Variable	1	2	3	4	5	6	7	8	9	10	11	12	13	14	15	16	17
1. PCS		0.01	−0.42 **	−0.43 **	0.05	−0.06	−0.10	−0.08	0.32 **	0.01	−0.17 ^†^	0.11	0.24 **	0.23 *	0.12	−0.20 *	0.03
2. MCS	0.00		−0.70 **	−0.69 **	−0.07	−0.09	−0.13	−0.23 *	0.27 **	−0.01	−0.28 **	0.09	0.17 ^†^	0.16 ^†^	0.06	−0.16 ^†^	−0.06
3. Depression	−0.38 **	−0.70 **		0.82 **	0.09	0.08	0.13	0.15	−0.49 **	0.06	0.39 **	−0.14 ^†^	−0.24 **	−0.30 **	−0.14 ^†^	0.17 ^†^	0.07
4. PTSD	−0.41 **	−0.68 **	0.82 **		0.15	0.13	0.23 *	0.27 *	−0.41 **	0.05	0.34 **	−0.14 ^†^	−0.24 **	−0.30 **	−0.13	0.24 **	0.10
5. DRDAY ^1^	0.01	−0.04	0.05	0.14		0.40 **	0.36 **	0.37 **	−0.04	0.03	0.08	−0.10	0.00	−0.11	−0.06	0.06	0.18 ^†^
6. AVEALC ^1^	−0.06	−0.06	0.05	0.10	0.28 **		0.53 **	0.57 **	−0.03	0.17	0.09	0.00	0.06	−0.17 ^†^	−0.05	0.14	0.07
7. HED ^1^	−0.10	−0.11	0.11	0.21 ^†^	0.37 **	0.56 **		0.56 **	−0.04	0.04	0.12	−0.11	−0.04	−0.15	0.02	0.05	0.07
8. HAZALC ^1^	−0.07	−0.22 *	0.12	0.24 **	0.39 **	0.66 **	0.56 **		−0.07	0.06	0.18 ^†^	−0.09	−0.01	−0.22 *	−0.14	0.16	0.08
9. Sleep quality	0.30 **	0.28 **	−0.47 **	−0.39 **	0.00	0.03	−0.04	−0.07		−0.07	−0.38 **	0.26 **	0.16 ^†^	0.11	0.09	−0.03	−0.24
10. Sleep duration	0.03	−0.05	0.03	0.02	−0.04	0.18 ^†^	0.03	0.01	−0.08		−0.01	0.26 **	−0.08	−0.04	−0.09	0.13	−0.01
11. Insomnia	−0.14 ^†^	−0.30 **	0.38 **	0.33 **	0.08	0.06	0.12	0.18 ^†^	−0.37 **	0.01		−0.22 *	−0.04	−0.12	−0.06	0.12	0.05
12. Rec. sleep	0.10	0.10	−0.15 ^†^	−0.14 ^†^	−0.09	0.01	−0.11	−0.09	0.26 **	0.16 ^†^	−0.21 *		−0.01	−0.03	0.03	0.03	0.19 *
13. PA: aerobic	0.24 **	0.17 **	−0.22 **	−0.24 **	−0.00	0.03	−0.04	−0.01	0.16 ^†^	−0.01	−0.03	−0.01		0.55 **	0.32 **	−0.13	−0.12
14. PA: strength	0.24 **	0.15 ^†^	−0.28 **	−0.29 **	−0.14	−0.15	−0.15	−0.22 *	0.11	0.01	−0.11	−0.03	0.55 **		0.46 **	−0.18 *	−0.02
15. PA: flexibility	0.11	0.07	−0.11	−0.12	−0.03	−0.10	0.02	−0.14	0.09	−0.07	−0.06	0.03	0.32 **	0.46 **		−0.01	0.07
16. Cigarette use	−0.20 *	−0.17 ^†^	0.18 *	0.23 **	0.11	0.11	0.05	0.16	−0.03	0.06	0.12	0.03	−0.13	−0.18 *	−0.01		0.18 *
17. Tobacco use	0.03	−0.05	0.07	0.09	0.23 *	0.05	0.07	0.08	−0.02	−0.05	0.05	0.19 *	−0.12	−0.02	0.07	0.18 *	

** *p* < 0.001; * *p* < 0.01; ^†^
*p* < 0.05. ^1^ Values reflect past month drinkers (72.1%). Note: Pearson correlation below diagonal; Spearman’s rho above diagonal; PCS = Physical Component Score; MCS = Mental Component Score; PTSD = posttraumatic stress disorder; DRDAY = drinking days; AVEALC = average drinks per drinking day; HED = heavy episodic drinking; HAZALC = hazardous alcohol use; Rec. sleep = recommended sleep; PA = physical activity.

## Data Availability

Restrictions apply to the availability of these data. Deidentified data may be available upon request and with the establishment of a Department of Defense data use agreement.

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
