# Peer review of "Long-Term Outcomes of Service Women Injured on Combat Deployment"

_ijerph, 2020, doi:10.3390/ijerph18010039_

Round 1
Reviewer 1 Report
Dear Authors,
I enjoyed reading your manuscript and am pleased to see these issues beginning to be introduced in the research literature. The following comments are offered in the spirit of scholarly feedback.
- Suggest the use of the construct, “sex-based” in place of “gender-based” and “biologically-based”. Consider framing your manuscript from a “sex-based” perspective and definition. The NIH Office of Research on Women’s Health (ORWH) further explains the differences:
"Sex" refers to biological differences between females and males, including chromosomes, sex organs, and endogenous hormonal profiles. "Gender" refers to socially constructed and enacted roles and behaviors which occur in a historical and cultural context and vary across societies and over time. All individuals act in many ways that fulfill the gender expectations of their society. With continuous interaction between sex and gender, health is determined by both biology and the expression of gender. (https://orwh.od.nih.gov/sex-gender)
- In the body of the paper, there are many examples of anthropomorphisms. Suggest the authors’ address.
- The authors have done a very nice job of describing each instrument. However, psychometric information about the instruments (e.g., SF-36, PTSD Checklist, PHQ-8, AUDIT, PSQI-7, RAPA), is lacking. Please update.
- Lines 243-245: Please consider that while this is important to operational readiness during active duty time, there are HUGE implications to service women if they continue active duty service or once they are medically retired due to these issues. And, while there are interventions (which you described) that can be used to help address these issues, our military health and Veteran mental health care systems are not making a big enough difference (some would argue little impact)--as evidenced by suicide rates (in both female and male service members who, for example, are medically retired due to sustained injury).
I look forward to seeing and reading your final publication in print~
Reviewer 2 Report
Title: ijerph-1042040- Long-Term Outcomes of Service Women Injured on Combat Deployment
Title
Does it clearly describe the article? - yes
Does it include the most important keywords (consider how you search for research articles) and demonstrate the significance of the research? - yes
Abstract
Does it accurately reflect the content of the article? - yes
Introduction
Good introduction with suitable sources.
Does this describe what the author hoped to achieve and clearly articulate the research question? Has the author provided a summary of the current research literature to provide context? - yes
Is it clear how this is being challenged or built upon? - yes
Are there any important works that have been omitted? – none identified
Methodology
Good description of the methods. It would be helpful to explain the relationship between the Defence Casualty Analysis System (line 65) and the Wounded Warrior Recovery Project (line 75) in order to determine the validity of the study sample of 230 service women from the entire cohort of injured service women. It would also help to explain whether this sample was solely naval personnel or covered service women from all services. Finally, it would be helpful to understand this sample of women as a proportion of the complete WWRP (which, I assume, includes men).
Does the author accurately explain how the data was collected? - yes
Is the design suitable for answering the question posed? Does the article outline the procedures followed? - yes
If the methods are new, are they explained in detail? – not required
Is there sufficient information present for you to replicate the research? - yes
Was the sampling appropriate? - yes
Have the equipment and materials been adequately described? - yes
Does the article make it clear what type of data was recorded; has the author been precise in describing measurements? yes
Results
Are results presented clearly? - yes
3.1. Demographics – the rationale for the paper is based on an assumed difference in health outcomes from injury on combat deployment between men and women - it would help to understand these demographics as a comparator to the males in the WWRP. It would also help to understand this cohort as a sample of all service-women exposed to combat during the study period.
3.2 and 3.3 – good summary
However there is no comparative analysis between demographic factors and health outcomes. This limits the interpretation of the results.
Discussion
Overall a good summary of the relevance of the results. However the male/female comparison with in the WWRP cohort should be supported by data that is presented in Tables within the results section as this whole dataset is owned by the study team. I believe that the implications of the results have been over-interpreted as a driver for additional clinical services – not every person with a ‘positive screen for mental ill-health in a questionnaire' meets the criteria of either a clinical diagnosis nor a requirement for active clinical intervention.
The paper states that 55.45% of the sample had left military service – this should be examined in the discussion.
Line 246 Paragraph 1 – useful scene setter.
Line 262 Paragraph 2/3 – this paragraph should be supported by including the summary data from the males in the whole WWRP cohort into Tables 1 and Table 2 so that the reader can actually confirm the assertions about comparability between genders in the WWRP cohort in paragraphs 2 and 3.
Line 285 Paragraphs 4/5 – this covers correlations from Table 3 – so should be specifically linked to the table. Given the number of comparisons, there is a statistical chance (greater than 0.05) of correlations between the variables that may not be clinically significant. Additionally, there are established causal relationships between the variables – e.g. mental health diagnosis and alcohol misuse, depression and poor sleep. The discussion might consider if the analysis adds any new insights into these relationships.
Line 306 the discussion on implications for healthcare providers makes a good case for holistic assessment of the health needs of this cohort of injured service women if they seek healthcare – but fails to differentiate between the use of questionnaire screening tools for assessment of measures of mental ill-health in sample populations and their validity in the clinical assessment of individual patients. The study provides no evidence for the assertion that the treatments listed in the sentence at 324 would have any impact in reducing the mental health scores at a population level in the questionnaires.
Conclusion
So you should consider are the claims in this section reasonable and supported by the results? - partially
Are the findings consistent with the author’s expectations? - partially
The paper opens by suggesting that there is a relative lack of knowledge about health outcomes in injured servicewomen compare to service men. The paper presents data on a cohort of servicewomen in the US Wounded Warrior Recovery Project. Whilst the discussion covers comparative differences with other studies, if does not compare the results with the male cohort in the WWRP. Therefore, the key results from this study cannot be fully interpreted. The other comparisons with ‘non-injured’ cohorts are interesting but self-evident. The positive correlation between combat injury and long-term positive responses to mental ill-health questionnaires has already been shown in multiple research programmes.
Language
Does the quality of English make it difficult to understand the author’s argument? – no
